# Nanoenergetic Composites with Fluoropolymers: Transition from Powders to Structures

**DOI:** 10.3390/molecules27196598

**Published:** 2022-10-05

**Authors:** Sreekumar Pisharath, Yew Jin Ong, Huey Hoon Hng

**Affiliations:** 1Emerging nanoscience Research Institute (EnRI), Nanyang Technological University, Singapore 639798, Singapore; 2School of Materials Science and Engineering, Nanyang Technological University, Singapore 639798, Singapore

**Keywords:** nanoenergetic materials, fluoropolymers, reactive structures, melt extrusion, casting

## Abstract

Over the years, nanoenergetic materials have attracted enormous research interest due to their overall better combustion characteristics compared to their micron-sized counterparts. Aluminum, boron, and their respective alloys are the most extensively studied nanoenergetic materials. The majority of the research work related to this topic is confined to the respective powders. However, for practical applications, the powders need to be consolidated into reactive structures. Processing the nanoenergetic materials with polymeric binders to prepare structured composites is a possible route for the conversion of powders to structures. Most of the binders, including the energetic ones, when mixed with nanoenergetic materials even in small quantities, adversely affects the ignitability and combustion performance of the corresponding composites. The passivating effect induced by the polymeric binder is considered unfavorable for ignitability. Fluoropolymers, with their ability to induce pre-ignition reactions with the nascent oxide shell around aluminum and boron, are recognized to sustain the ignitability of the composites. Initial research efforts have been focused on surface functionalizing approaches using fluoropolymers to activate them further for energy release, and to improve the safety and storage properties. With the combined advent of more advanced chemistry and manufacturing techniques, fluoropolymers are recently being investigated as binders to process nanoenergetic materials to reactive structures. This review focuses on the major research developments in this area that have significantly assisted in the transitioning of nanoenergetic powders to structures using fluoropolymers as binders.

## 1. Introduction

The concept of structural nanoenergetic materials (SNE) is an emerging area of interest in the field of reactive materials. Research strives to develop composite materials that can serve as high strength structural materials and at the same time can be controllably stimulated to release energy on demand. These materials consist of hard, non-dissipative fillers of reactive materials dispersed in a polymer binder matrix. The reactive material reinforcements mostly comprise of high enthalpy components that include metal fuels, intermetallic alloys, and thermite powder mixtures. The polymer binder in the formulation plays the critical roles of providing the beneficial processability, safety and mechanical properties for the SNE composites.

Inert binders such as epoxy polymers, which do not contain energy releasing groups, have been previously investigated as binders for SNEs [1]. Even though the binders did meet the critical desired mechanical properties, the performance characteristics deteriorated due to the energy diluent effect of the inert binder system.

Energetic binders with oxidizing or energy releasing groups are expected to provide advantages of: (a) higher density than their hydrocarbon counterparts; (b) additional oxidizing capability that will enhance the performance; (c) manipulating the mechanical properties of the composite. Furthermore, energetic binder formulations require lower solid loading of sensitive fillers compared to the non-energetic binder ones to achieve similar performance levels. Lower sensitive filler loadings translate to reduced vulnerability of the composites towards external stimuli and, therefore, greatly increase the safety of the composites containing them [2].

Energetic polymer binders such as glycidyl azide polymer (GAP) and nitrocellulose (NC) have been successfully used as binders for SNE composites intended for thruster applications by taking advantage of their excellent gas generating properties [3]. However, such energetic binders have been reported to deteriorate the ignitability of the SNE composites, especially at higher binder contents [4]. Additionally, these energetic binders are unable to meet the demanding mechanical requirements of the SNE composites, specifically elastic and compliant nature over rapid temperature changes. Therefore, the application of these energetic binders to SNE are restricted to mostly coating applications, where the amount of energetic binder will not be higher than 10 wt% with respect to the composite. Consequently, the SNEs were studied in the form of coated powders, as the amount of energetic binder is inadequate to form SNE-based reactive structures [5].

Fluoropolymers possess excellent properties, such as outstanding chemical resistance, high thermal stability, low coefficient of friction, and diverse mechanical properties [6]. These properties come from the special electronic structure of the fluorine atom, the stable carbon-fluorine covalent bonding, and the unique intramolecular and intermolecular interactions between the fluorinated polymer segments and the main chains [7]. The presence of oxidizing fluorine atom provides the necessary energetic properties for the binder system [8]. Specifically, with reference to the fluoropolymer composites containing metal fuel, aluminum (Al), the oxidizing fluorinated fragments generated from the decomposition of fluoropolymer induces pre-ignition reactions (PIR) with the nascent alumina shell [9,10]. The PIR paves the way for the full-fledged combustion reaction of Al and sustains ignitability of the composites even at higher polymer binder contents [11]. As a result, the emergence of fluoropolymers as binders has opened the possibility of transitioning the SNE composites from coated powders to mechanically robust reactive structures. Therefore, over the years, fluorinated polymers have received significant interest as energetic binders and have dominated the research into SNE composites [12,13]. In this light, it is instructive to review the development in this area of research.

Over the years, a wide range of processing methods, from conventional methods such as pressing, extrusion and cast curing to modern additive manufacturing (AM) technologies, have been adopted to produce the fluoropolymer-based reactive structures [14]. The cure chemistry and the physical properties of fluoropolymer-based binders have been suitably developed or manipulated to cater to these diverse processing requirements. This review will highlight the progress made in the use of fluoropolymer-based energetic binders in the research of SNE composites over the last decade, dealing predominantly with the production of reactive structures. The reactive materials discussed in this review will be restricted to compositions with Al as the fuel component, as they are the most widely investigated and much sought-after among the nanoenergetic materials for practical applications. This review will address the following aspects of the research on SNE composites: (a) Energetics of aluminum oxidation with and without a fluoropolymer. (b) Reactive structures based on fluoropolymer-based binders. (c) Future directions.

## 2. Energetics of Aluminum Oxidation with and without a Fluoropolymer

A basic understanding of the energetics of Al oxidation in the absence and presence of fluorine containing polymers is essential to appreciate the utility of the fluoropolymer binders to produce the Al-based reactive structures. The physical and thermochemical properties of Al_2_O_3_ and AlF_3_, which are the primary combustion products of Al oxidation and fluorination, are compared in Table 1 [15,16].

As seen in Table 1, the heat of formation of AlF_3_ considerably exceeds that of Al_2_O_3_. Accordingly, the energy released during Al fluorination (−56.1 kJ/g) is nearly twice that liberated during oxidation (−30.98 kJ/g). Furthermore, the physical properties of Al_2_O_3_ and AlF_3_ are quite different. With respect to the boiling point of Al (2470 °C), Al_2_O_3_ evaporates at a higher temperature of 2977 °C, but AlF_3_ sublimes at a much lower temperature of 1275 °C. The stark differences in the thermochemical and physical properties of aluminum oxide and fluoride would lead to operation of diverse mechanisms of the oxidation of Al in the absence and presence of the fluoropolymer coating (Figure 1).

Without a fluoropolymer, because the vaporization temperature of Al_2_O_3_ is higher than the boiling temperature of Al, the molten Al must first diffuse through the dense and refractory oxide layer to react with the oxygen in the vapor phase to undergo combustion. The combustion reaction rate will further slow down with the increase in the thickness of the oxide layer as the reaction proceeds. This limits the complete utilization of the enthalpic potential of Al resulting in longer ignition delays and low combustion efficiencies.

In the presence of a fluoropolymer, the oxidizing fluorinated species formed from the decomposition of the fluoropolymer first reacts with the Al_2_O_3_ shell to form AlF_3_. This is known as the pre-ignition reaction (PIR). The AlF_3_ thus formed sublimes at 1275 °C, which is well below the vaporization temperature of Al. Sublimation of AlF_3_ results in the removal of the diffusion limiting Al_2_O_3_ shell and exposes fresh Al surface to combustion, and thereby enhancing the overall kinetics of the reaction. The activating effect of the fluoropolymers as shown in Figure 1 has been experimentally confirmed by Nie et al. [17], in which the fluoropolymer coated Al and Al-Mg alloy powders exhibited improved combustion reactivity in the constant volume explosion test and recorded higher combustion temperatures as compared to the respective pristine powders. Therefore, fluoropolymers have been established as promising binders for Al-based reactive structures.

## 3. Reactive Structures Based on Fluoropolymer-Based Binders

### 3.1. Composites with Commercial Fluoropolymer Binders

Historically, the application of fluorinated polymers in reactive material formulations started with the introduction of polytetrafluoroethylene (PTFE) into the commercial markets in 1946 [18]. The crystalline polymer PTFE (Figure 2), has distinguishing properties such as higher thermal stability (~350 °C), density, oxidizing capability (fluorine content ~72%) and good mechanical properties over the widest temperature range known for any polymer.

The earliest fluoropolymer-based reactive material to be developed and investigated was PTFE/magnesium (Mg) mixtures, in which PTFE served as an oxidizer or fluorine source for the reaction with Mg metal fuel. Over the past decades, PTFE mixtures with alternate metal fuels were developed. Among them, PTFE/Al mixtures have received significant attention as a reactive material formulation [19]. Aluminum nanoparticles, on account of their higher reactivity [20], have been progressively used as metal fuels in PTFE formulations as compared to the conventional micron-Al powders.

When the PTFE/Al composite is subjected to a strong mechanical stimulus such as impact, Al undergoes highly exothermic oxidation reaction using the reactive fluorinated species generated from the decomposition of PTFE. The most widely used PTFE/Al formulation is the stoichiometric composition containing 73.5 wt% PTFE and 26.5 wt% Al [21]. The reaction releases 8.7 kJ/g of heat. As previously stated, the oxidation reaction is triggered by the pre-ignition reaction (PIR) between the alumina shell and the fluorinated species resulting in the gasification of alumina shell and formation of the AlF_3_. The exothermic formation of AlF_3_ produces combustion temperatures in the order of 3000 °C that is adequate to melt and sublime the formed AlF_3_, thus exposing fresh Al core to further fluorination reaction. Consequently, PTFE/Al composites undergo self-sustaining combustion reactions under dynamic loading conditions which is a key requirement for the reactive structures. In that sense, PTFE/Al formulation serves as a benchmark for reactive material development.

The energy release and mechanical properties of the stoichiometric formulation of PTFE/Al composite have been investigated extensively [22]. It has been concluded that the impact initiation potential of the composite is closely related to the dynamic deformation behavior of the composites. The deformation behavior of PTFE/Al composite material has been largely characterized as elastoplastic in nature. Under dynamic loading conditions, the composite has been observed to undergo extensive strain hardening with flow stresses increasing linearly with the strain. Additionally, the composite also exhibits strain rate hardening effects, with the hardening modulus and compressive strength showing a positive correlation with the strain rate. At higher temperatures, PTFE undergoes significant thermal softening with the compression strength of the composite decreasing by almost 50% when the composite temperature increased from 20 °C to 200 °C [23]. 

Microstructurally, at high strain rates, the PTFE matrix is drawn into fine fibrils of diameter as thin as 60–100 nm, which bridge the opposite faces of the interfacial crack providing additional resistance for its propagation. The fibrils are formed because of extensive crazing of the PTFE matrix under applied load. Thus, in the PTFE/Al composite the PTFE polymer plays a dual role of strengthening and toughening of the composite. The PTFE/Al composite exhibit a gradual change from brittleness to ductility with increasing temperature due to the enhanced PTFE deformation [24]. Remarkably, the minimum specific incident energy that causes ignition of the composite reduces by 58% when the temperature is raised from 22 °C to 205 °C. The reduction has been attributed to thermal softening of the PTFE matrix that has presumably resulted in the generation of large number of critical ignition hotspots [25]. Mechanistically, the critical ignition hotspots are generated because of the formation of discrete adiabatic shear bands by virtue of the thermal softening of the PTFE [26].

Sintering is the key processing parameter involved in the production of the PTFE/Al composites. The sintering temperature directly influences the mechanical and energetic responses of the composites. For example, the PTFE/Al composites sintered at 310 °C underwent brittle failure under quasi-static compression and did not undergo any reaction [27]. On the other hand, specimens sintered at 360 °C exhibited remarkable toughness and reacted almost completely [27]. The enhanced toughness assists in the absorption of more energy before failure and initiates the combustion reaction. The unreacted specimens showed strain softening, whereas the reacted specimens demonstrated pronounced strain hardening after yielding [24]. The change in mechanical behavior has been attributed to the variation in crystallinity of the PTFE matrix with respect to the sintering temperature. Below 310 °C, the crystallinity of PTFE is as high as 90%. With increasing sintering temperature, the crystalline phase of the PTFE transforms into the amorphous phase, that melts together to form a co-continuous phase. At the molecular level, the enhanced toughness of the more amorphous PTFE phase has been ascribed to the faster orientation of the chain entanglement regions with respect to the crack propagation speed that would act as a toughening mechanism [28].

The addition of extra additives, such as tungsten (W) particles, was explored to further improve the density, lowering the collateral damage of the composite and probable participation in the exothermic reactions of the Al/PTFE composites. Cai et al. [29] carried out an investigation into the quasi-static and high strain rate mechanical properties of composites consisting of Al, PTFE, and W particles by testing the samples with different sizes of W particles. It was observed that porous composite samples containing fine W particles have higher quasi-static and dynamic fracture stresses than their higher density counterparts containing coarse W particles. This unusual feature was explained by the shifting of the deformation behavior from elastoplastic with significant strain hardening to being brittle with no hardening. The research in this area is further widened with the assessment of other additives such as nickel (Ni) [30], copper oxide (CuO) [31], zirconium and titanium hydrides [32,33], and silicon carbide (SiC) [34], with the objective of further improving the mechanical and reactivity properties of Al/PTFE composites.

The potential military applications of PTFE/Al SNE composites are as reactive fragments, and structural components in warheads, shaped charge liners etc. More extensive application of PTFE as a binder for SNE composites is hindered by the high melt viscosity of PTFE (10^11^ Pa·s at 380 °C). This constrains the viable processing operations for the manufacturing of SNE composites to the expensive and energy intensive sintering. Furthermore, the sintering temperature of ~360 °C required to achieve the best mechanical and energetic response for the PTFE/Al composites is very close to the autoignition temperature of the formulation. Therefore, sintering would pose serious safety risks for the processing of the PTFE/Al composites. Lack of melt processability is a major hindrance for the high throughput production of the PTFE-based SNE composites.

Over the years, research on easily processable fluoropolymers has led to the commercial development of several potential candidate binders for reactive materials with improved processability attributes. The key candidates include poly(vinylidene fluoride) (PVDF), Viton^®^ (copolymer of vinylidene fluoride (VDF) and hexafluoropropene (HFP)) and THV (copolymer of tetrafluoroethylene (TFE), HFP and VDF). The structures of the polymers are shown in Figure 3 and the key properties are summarized in Table 2 [7].

In addition to the well-known commercial fluoropolymer binder candidates listed in Table 2 that are amenable to melt extrusion and solvent casting processing operations, non-commercial cast curable fluorinated binders have also been reported. Development of these binders have revealed newer possibilities to manufacture reactive structures. In the light of this, it is instructive to briefly elaborate on the melt extrusion, solvent casting, and cast curing operations, that have been used in the manufacturing process.

In melt extrusion, the molten thermoplastic binder and the reactive ingredients are extruded with a rotating screw under elevated temperature through a die into a structure of uniform shape. The intense mixing caused by the screw assists in the uniform dispersion of the ingredients in the polymer matrix. Melt extrusion operates continuously thereby reducing the number of involved unit operations and improves the economic viability of the reactive structure manufacturing. Nevertheless, the quality and performance of reactive structures processed through melt extrusion are limited by the friction and thermal sensitivities of the reactive ingredients in the formulation. In solvent casting, binders that are soluble in volatile solvents are used to disperse the reactive ingredients at a reasonable solid concentration. The dispersion is casted into a mold and the solvent is then evaporated to obtain the requisite shape. The viscosity of the dispersion is a critical parameter that determines the quality, size, and performance of the reactive structure. One key challenge is the residual porosity in the structures caused by evaporation of the solvent, which can be tackled by the application of vacuum. In cast curing, the reactive ingredients are mixed with a liquid prepolymer and casted into a mold. The mixture is then converted into a permanent polymeric binder bound composite by reacting with a suitable curing reagent. Conventionally, the employed cure chemistry is based upon polyurethane linkages formed by the reaction of the hydroxyl-terminated liquid prepolymer and multifunctional isocyanate curing reagent. The key process parameters to be considered are (a) ability of the binder to wet solids, (b) final mix viscosity, and (c) flowability of the mix into the mold.

The following sections detail examples of reactive structures processed through the melt extrusion, solvent casting, and cast curing manufacturing operations.

#### 3.1.1. Reactive Structures through Melt Extrusion

As compared to PTFE, although lower in fluorine content, the development of easily processable binders presented in Table 2 provided a paradigm shift to the processing methodologies adopted for producing fluoropolymer-based reactive structures. From the energy intensive and expensive sintering process required to produce the PTFE/Al composites, the processing technology shifted to more economical and higher throughput melt extrusion/injection molding methods, especially for Al composites with THV and PVDF polymers.

McCollum et al. [35] prepared nano-Al and PVDF composites by injection molding and investigated the morphological and combustion properties of the prepared composites (Figure 4).

In this work, the nano-Al loading in the composite formulations was varied from 0 to 8 wt%. The pre-mixed dry powders of nano-Al and PVDF were melt compounded at 190 °C and 250 rpm for 10 min before injection molding into coupons (Figure 4a). The nano-Al particles were observed to be well dispersed in the PVDF matrix even at the highest concentration of nano-Al used in the formulation (Figure 4b). Little to no interaction existed between Al particles and the PVDF chains as indicated by the unvarying key transitions of the PVDF in the DMA signatures of the composite coupons (Figure 4c). Composites that were loaded with 6 and 8 wt% Al could be ignited by a propane torch to self-sustained combustion at a linear burn rate of 3 and 4 mm/s (Figure 4d). The primary reaction product AlF_3_ was observed in the post-burn residues of the combusted samples, thereby producing the most energetic response driven by the reaction between the alumina layer and hydrogen fluoride formed from the catalyzed decomposition of PVDF. Apparently, the fluorination reaction is promoted by the strong interface between Al and PVDF in the composite coupon. The key challenge faced in this work was the decline in the processability of the samples with increasing Al concentration as indicated by a steady decrease in the melt flow rate which prevented higher loading of nano-Al powders into the composite. Despite the processability problem, this work opened the possibility of adaptation of the Al/PVDF composite materials to melt processing-based additive manufacturing (AM) technique such as fused deposition modelling (FDM) method.

In a continuing work from the same group [36], polymethylmethacrylate (PMMA) was added to the Al/PVDF formulation with the objective to improve the processability and adhesion properties while adopting the FDM manufacturing method (Figure 5). The pre-hand mixed components of the formulation were melt compounded at 190 °C and extruded to form filaments of approximately 2 mm in diameter and 0.5 m in diameter. The filaments were in turn fed to a 3D printer that printed the formulation onto a pre-heated glass plate through a nozzle heated at 230 °C.

The melt flow rate of the formulation was observed to increase with increasing PMMA concentration indicative of the improved processability. This was attributed to the miscibility of the PVDF and PMMA phases. Because of the improved processability, the loading of the nano-Al could be further increased to a maximum of 30 wt%. Due to the inert nature of PMMA, the concentration of PMMA was restricted to a maximum of 25 wt%. Interestingly, the burn rate of the samples showed opposing trends in the fuel lean and fuel rich samples with respect to the increasing PMMA concentration. The burn rates of the fuel lean composites containing 15 wt% nano-Al increased from 2 to 4.4 cm/s when the PMMA concentration increased from 0 to 25%. Whereas, the burn rate of the fuel rich composites containing 30 wt% nano-Al decreased from 13 to 4.1 cm/s in the equivalent concentration range of PMMA. The observed trends were attributed to the changes in the stoichiometry of fuel/oxidizer mixtures, with the composite containing 70% PVDF, 30% nano-Al and 0% PMMA exhibiting the best combustion rate properties. Like the previous study, α-AlF_3_ was formed as the primary combustion by-product suggesting that the combustion chemistry between nano-Al and PVDF is unaffected by the PMMA addition to the formulation. It was concluded that the addition of PMMA assisted in adapting the Al/PVDF formulation to the FDM technique and tailored the combustion performance of the composite to suit specific requirements.

In an optimization effort, the processability and combustion behavior of the melt processed Al/PVDF composite was investigated as a function of the molecular weight of PVDF [37], the fluorine content of PVDF and Al particle size [38]. Processability of the unfilled and fuel lean (15 wt% Al) formulations was observed to be improved with a reduction in molecular weight of PVDF. However, this improvement was curtailed in fuel rich formulations containing 30 wt% Al due to increased electrostatic interaction between nano-Al and the PVDF chains. The combustion reactivity of the composites was observed to decrease with decrease in the molecular weight of the PVDF. The lower reactivity has been ascribed to the diffusion barrier created by the low molecular weight PVDF chains near the Al particle surface resulting in negligible formation of the AlF_3_ combustion product. At a constant molecular weight, increased fluorine content of the PVDF binder (the fluorine content of the binder was altered by varying the hexafluoropropylene (HFP) content in PVDF) resulted in higher consumption of Al driven by the enhanced fluorination reaction. The consumption of Al increased with decreasing particle size of Al due to the increased fuel surface area. In other words, under comparative loadings, more Al remains unreacted in the Al/PVDF composite samples containing Al with larger diameters (10 μm).

In a related work, Fleck et al. [39] demonstrated the feasibility of FDM to print Al/PVDF filaments containing 20 wt% micron-scale Al particles. In their work, the melt extruded filaments were fed to a Makerbot Replicator 2X 3D printer operating at nozzle temperature of 230 °C to obtain the printed filaments. Upon ignition using nichrome wire, 3-inch diameter sections of the printed filaments combusted with an average burn rate of 1.5 cm/s (Figure 6a). The burn rate property showed good batch to batch repeatability. A key achievement of this work was the standardization of the print quality of the energetic filaments with that of standard materials such as acrylonitrile butadiene styrene (ABS) using bead to bead adhesion and surface quality as metrics. As shown in Figure 6b, the quality of the printed reactive structure was found to be comparable with that printed with the standard material.

After establishing the quality of the print, FDM was further used to print ASTM D638 Type V dog bone specimens of the composites (Figure 7) [40].

The mechanical and combustion properties of the printed composites were investigated as a function of micron scale Al loading (varied from 10 wt% to 50 wt% in 10 wt% increments; particle size of Al kept constant at 4.5 μm) and particle size. The key objective of the work was to determine the influence of material selection on the structural and energetic properties of the 3D printed reactive structures. The modulus of elasticity increased from 71 MPa to 208 MPa when the Al particle content was increased from 10 wt% to 50 wt%. However, the tensile strength of the composites recorded a 37% reduction from 34.5 MPa to 21.6 MPa over the same micron-Al loading range, most likely due to poor interfacial interaction between Al and PVDF matrix.

Thermochemical code (Cheetah 7.0) was used to calculate the theoretical heat of combustion and adiabatic flame temperature of the composites. The theoretical heat of combustion increased linearly with increasing Al content as Al releases more energy during combustion in air compared to PVDF. The adiabatic flame temperature shows a peak value at 20 wt% Al that was closer to the stoichiometric ratio of Al to PVDF. The experimentally determined burn rate of the composite samples showed a maximum value (~25 mm/s) at 30 wt% Al loading, which was just above the stoichiometric conditions for the Al/PVDF system.

With reference to the variation in particle size, where the particle size variation was limited to the 4.5 μm to 30 μm range, no significant trend was observed in either the elastic modulus or tensile strength. However, at a constant Al loading of 20 wt%, the burn rate of the reactive structures showed a decreasing trend with respect to increasing particle size due to the decrease in effective surface area for the fuel to react with the oxidizer.

The tensile properties of the printed Al/PVDF reactive structures were also investigated as a function of the bead orientation by printing the composites in 0 deg and 90 deg extrusion directions [41]. It was found that, irrespective of the particle loading, 0 deg orientation resulted in the highest modulus of elasticity and highest ultimate strength. A reduction of 5% was recorded for the ultimate strength when the print direction was shifted from 0 deg to 90 deg. The ultimate tensile strength reduced in higher magnitude of 20% with respect to the change in the print direction from 0 deg to 90 deg. Interestingly, under similar conditions, the mode of failure of the composite also changed from necking or yielding, which is more characteristic of the deformation of the polymer, to a tensile fracture determined by the interface between the Al and PVDF. No distinguishable difference in burn rates of the composites was observed with respect to the orientation of the printing, with both the composites registering a burn rate value of ~21 mm/s. To simulate a realistic structure design, two trusses with different internal designs A and B were printed (printing done with comparable parameters) and tested under flexural load using a 3-point bend fixture. The trusses were observed to perform differently, the one with internal design B providing higher flexural stress and strain as compared to A. Thus, the potential to control the structural performance during the conception of the design of the reactive structure was demonstrated.

Taking advantage of the tailorable processing attribute of the FDM technique, Hoganson et al. [42] printed stoichiometric micron-Al/PVDF reactive structures of varying densities and in-fill geometries and measured their energy release properties in terms of the burn rate. The structures were printed in concentric, gyroid, grid, and a customized tortured path geometry, and filled to different in-fill percentages of 0%, 30%, 75%, and 100%. Every structure was printed with a constant exterior geometry of 5 × 5 × 30 mm, with an additional 1 mm at the top for an igniter seating region to ensure homogeneous ignition conditions across all samples (Figure 8). Ashby plots of density versus energy release rate were generated from the measurements for each of the structures (Figure 8).

Across the shapes printed, 75% in-fill samples consistently generated higher energy release rates and the corresponding 100% in-fill samples demonstrated lower energy release rates. Although gyroidal infills showed a slight tendency to have higher energy release rates than concentric infills, in general the energy release rates were not found to be a strong function of the density and the in-fill shapes.

Collard et al. [43] investigated the combustion properties of melt extruded reactive filaments consisting of 20 wt% nano-Al/PVDF and 32.2 wt% mechanically activated (MA) Al-PTFE/PVDF. Quantitative porosity analysis of the extruded filaments was conducted using X-ray microcomputed tomography (MicroCT). The total porosity of the nano-Al/PVDF composite filament was observed to be 11.49% and that of the MA Al-PTFE/PVDF composite was 19.05%. The average burning rate of the nano-Al/PVDF composite filament (average diameter = 1.51 mm) was 40.6 ± 2.5 mm/s and that of the MA Al-PTFE/PVDF composite filament (average diameter = 1.49 mm) was 24.3 ± 1.4 mm/s. The flame propagation rates were observed to be linear, indicating that the filament material had a homogeneous distribution. The burn rates reported in this work are higher than that reported by Fleck et al. [39] for the composites with micron-Al, thus clearly demonstrating the advantage offered by nano-Al for improving the combustion performance. This work convincingly established the feasibility to safely melt process nano-Al containing reactive structures which is considered as a significant research advancement in this area. However, much research remains to be done to address the challenges of maintaining reasonable porosity, scaling to larger sample sizes, and minimizing the defects of the reactive structures.

#### 3.1.2. Reactive Structures through Solvent Casting

In addition to the melt processability, the solvent solubility property of PVDF, THV, and Viton^®^ binders were utilized for the direct writing of reactive structures containing Al fuel. Wang et al. [44] used solvent-based direct writing to prepare free standing reactive films of nano-Al with PVDF, THV, and Viton^®^. In this work, the homogenized precursor solutions containing the respective fluoropolymer and fuel mixture was printed through a 700 μm nozzle onto a pre-heated glass slide kept at 80 °C at the rate of ~4.5 mL/h. Composite films containing 10 layers were prepared with a fixed polymer concentration and a different mass fraction of nano-Al so that the composition of each film was closer to the stoichiometric ratio. The composite films were further characterized for mechanical, ignition, and combustion properties. Among the films, the Al/PVDF film was able to support the highest tensile strength of ~35 MPa, followed by the THV (~6.5 MPa) and the Viton^®^ (~1.2 MPa). The highest tensile strength of the PVDF film was ascribed to the unique honeycomb like microstructure formed during the printing of the film. The Al/Viton^®^ film could endure >30 times higher strain before fracture due to the elastic nature of Viton^®^.

The burn rate and combustion temperature generated by the composite films were dependent on the nature of the fluorinated species generated by the pyrolysis of the respective fluorinated polymer, which was inherently related to its fluorine content. The PVDF polymer with ~59% fluorine content generates predominantly hydrogen fluoride (HF) during pyrolysis that triggers pre-ignition reaction with alumina layer of the nano-Al. This resulted in the lowest ignition temperature for the Al/PVDF film and thus apparently the fastest burn rate (Figure 9). On the other hand, THV, the polymer with the highest fluorine content of ~73%, releases more CF_x_ species under pyrolysis, and thus producing the highest combustion temperature of ~2500 K among the films.

In addition to Al/PVDF, reactive films based on PVDF, and Al-based nanothermites have also been investigated for mechanical, ignition and combustion properties [45]. The nanothermite compositions used in the study were mixtures of nano-Al with metal oxides such as bismuth oxide (Bi_2_O_3_), copper oxide (CuO), and iron oxide (Fe_2_O_3_). The films, with nanothermite loadings varying from 50 wt% to a maximum of 67 wt% with respect to the PVDF binder, were prepared by the solvent-based direct ink writing technique. The simultaneous presence of two oxidizing moieties such as the metal oxide and the PVDF binder provides a complex mechanistic environment for the oxidation of Al. Therefore, to avoid the presence of a third oxidizer in the form of the air, the ignition and the combustion studies in this work were done in either a vacuum or an argon environment.

Mechanical integrity of the films was observed to be reduced with increasing loading of the nanothermites into the PVDF due to the formation of pores in the films. The reactive films showed an overall trend of a reduction to the energy release rates with the inclusion of metal oxides. However, the flame temperatures generated by the films increases with increase in the nanothermite loading. Among the composites, those containing CuO burn the hottest with temperatures nearing ~2500 K, especially at higher mass loadings. Those incorporating Bi_2_O_3_ and Fe_2_O_3_ burned at temperatures between 1400 K to 1600 K. Probing the reaction mechanism indicated that Al/PVDF reaction remains the key for the ignition of the composite film due to pre-ignition reaction between the HF generated from PVDF and the encapsulating alumina layer of the nano-Al. The added metal oxides were unable to synergistically contribute to the overall kinetics of the Al oxidation due to the side reaction of the metal oxide with the carbon species generated from the decomposition of the PVDF. This, in combination with the loss of mechanical integrity of the films due to the gas generation, adversely affected the combustion performance.

The results from this work demonstrated that mere mixing of the metal fuel with PVDF to form a reactive structure would not yield promising results. Instead, it is necessary to control the stoichiometric balance and microstructure of the films to obtain optimal energetic and mechanical performance from the reactive structure.

The solvent solubility of F2311 fluoropolymer, a copolymer of chlorotrifluoroethylene and vinylidene fluoride was also exploited to obtain 3D printed core-shell reactive structure based on Al/CuO nanothermite [46]. Rheological investigation of the formulation with 15 wt% F2311 was found to exhibit promising printing performance. Based on the rheology result, nanothermite ink containing 15 wt% F2311 was prepared in butyl acetate solvent and printed through a customized nozzle to obtain a core-shell reactive structure (Figure 10b).

The nanothermite hollow fiber with diameter 0.9 mm and length 10 cm was ignited using a nichrome wire and the flame propagation was observed using a high-speed camera. Interestingly, the authors observed a remarkable difference in the combustion behaviors with respect to the geometry of the reactive structure. The hollow fiber recorded a significantly higher flame propagation rate of 395 m/s as against 0.09 m/s for the solid fiber. The burn rate of the fibers decreased with increasing fluoropolymer content and vice versa. The higher flame propagation rate of the hollow fiber reactive structure was attributed to the “cavity mediated self-accelerating combustion” mechanism, in which the heat transfer to the propagating front was presumed to be promoted by the cavity of the hollow fiber architecture. Similar F2311-based hollow reactive structures have been reported with other Al-based nanothermites, and reactive materials were formulated with Fe_2_O_3_, Bi_2_O_3_ and PTFE as oxidizers [47].

Zheng et al. [48] investigated the thermal and combustion properties of the 3D printed reactive structures consisting of nano-Al/PTFE powders and F2311 fluoropolymer. In this work, the reactive ink was prepared by mixing the appropriate amounts of pre-mixed nano-Al and PTFE powders with F2311 in butyl acetate solvent. Ink formulation with 15 wt% F2311 content, found to be the most stable for printing, was printed on multiple layers into a 3D mesh type reactive structure using a 0.21 mm diameter needle. The ink formulation was further printed into 4 cm long sticks for burn rate measurements (Figure 11). The burn rates of the composites were studied as a function of the diameter of sticks (varied from 0.41 mm to 1.06 mm) and the Al/PTFE ratio (varied from 0.3 to 1.8) in the reactive ink formulation by keeping the F2311 content constant at 15 wt%. Even though the flame size increased with increasing diameter of the sticks, the burn rates of the sticks were observed to be constant. With increasing the Al to PTFE ratio in the ink formulations from 0.3 to 1.8, the burn rates of the sticks increased from ~2 cm/s to 10 cm/s. 

The above works clearly demonstrated the versatility of the fluorinated binders such as PVDF to fabricate reactive structures by using economical and high throughput techniques. However, as seen in the presented examples, the sizes of fabricated reactive structures are limited to a maximum of centimeters, which is inadequate for energetic material applications that require macroscale features. Further scaling of the size of the reactive structures is mostly limited by the safety concerns as well as the challenge on maintaining the homogeneity of the composites. Therefore, alternate fluoropolymer-based binder systems needed to be developed that would assist in adapting the processing of the reactive structures amenable to conventional polymer processing techniques such as melt extrusion.

### 3.2. Alternate Approach to Reactive Structures through Melt Extrusion

Crouse et al. [49,50] reported a newer approach towards the preparation of nano-Al/fluoropolymer composite-based reactive structures in which the nano-Al was chemically integrated into a melt processable fluorinated methacrylic polymer matrix. This approach would ensure good interfacial interaction of the nano-Al and the fluorinated matrix that would enhance the mechanical properties of the reactive structure without compromising on the reactivity. In this work, the aluminized fluorinated acrylate (AlFA) composite was prepared by in situ free radical polymerization of acrylic functionalized nano-Al particles with a commercially available fluorinated methacrylate. The approximate fluorine content of the fluorinated methacrylate was ~60%, closer to that of PVDF. The composite was further compounded in a bench-top twin-screw extruder at 150 °C and extruded through a circular die into a copper-clam shell mold (Figure 12a). The size of the extruded composites was 4 mm diameter × 8 mm long. Theoretical maximum densities (TMDs) were calculated for the 4 mm diameter × 8 mm long pellets and ranged from 99% for the neat polymer to 88% and 94% TMD for the AlFA-50 and AlFA-60 composites, respectively. Thermoplastic behavior enabled the composites to be processed by extrusion. The processing temperature was lower than that used for the extrusion of PVDF-based formulations. Notably, composites with nano-Al loadings up to 60 wt% or less could be successfully prepared by extrusion.

As seen in Figure 12b, in contrast to the result reported by Fleck et al. [40] for the Al/PVDF composites, the compressive strength of the AlFA composites increased from 11 MPa to 28 MPa with the addition of 60 wt% nano-Al. The compression test results vividly demonstrated that the chemical integration is a viable strategy for improving the interfacial interaction of nano-Al and the binder, and thus enhancing the mechanical properties. 

Open air combustion tests of the AlFA composites indicated that all the composites except those containing 0 and 10 wt% nano-Al demonstrated self-sustaining combustion upon ignition by a butane torch. The composites prepared with 50 wt% nano-Al particles demonstrated the highest reactivity with the combustion reaction proceeding by both oxidation and fluorination mechanisms.

Using a modified Taylor’s gas gun test set up, White et al. [51] further investigated the impact initiation of the AlFA composites with 50 wt% nano-Al (Figure 13a). The investigation was carried out by impacting the composite pellets (3.5 mm diameter × 1.5 mm long) against a steel or sapphire anvil at a nominal velocity of 150 m/s. The ignition process was observed using the side profile as well as the head on images acquired using a high-speed camera.

The first light from the ignition was observed in the range of 7 to 10 μs, when the AlFA composite pellet appeared to be fully compressed and extensively deformed (Figure 13b). The AlFA composite sample was able to endure a large amount of strain without undergoing fracturing or fragmentation due to the high polymer matrix content (~50 wt%). Reaction was initiated by the strain induced mixing of the nano-Al with the fluorinated matrix in the presence of the oxygen from the air. The ignition location, as revealed by the head on images (Figure 13c), was located at an intermediate radius within the compacted composite. Multiple ignition locations, that were arranged in a crescent shape, were observed. Within 2–3 μs after the ignition, the reaction progress was observed as a uniform circular ring that expands radially as a function of time.

The AlFA composite deformation was described using a Zerilli–Armstrong model that was derived by using the experimental Al-PTFE literature data. The model parameters were further used in ALE3D continuum simulations to predict the shear induced ignition locations in the composite, which correlated well with the experimentally observations. The simulations indicated that the impact initiation was driven by the generation of shear bands, a known mechanism for the impact initiation of energetic materials, formed because of the accumulation of a large number of dislocations along the slip planes of the impacted solid. The buildup and release of dislocations could occur in rapid succession, releasing large amounts of localized energy forming shear bands and causing material ignition. The adiabatic shear band generation was aided by the complete deformation of the impacting copper projectile inside the AlFA composite pellet. The temperatures within the shear band were sufficiently high to initiate the decomposition process resulting in the ignition of the composite.

Research efforts have also been undertaken to tailor the sensitivity of the AlFA composite to impact ignition for a broader range of scenarios by sensitization approaches [52]. It was noted that the AlFA composite could be ignited at impact velocities as low as 35 m/s by the addition of 15% glass micro-balloons by volume. The addition of the 15% glass micro-balloons creates effective "pinch points" that would result in the viscous heating and consequent ignition of the composite. The level of material sensitization could be effectively altered by varying the concentration of the glass micro-balloons in the range of 10 to 30%. This observation opens the possibility for the AlFA composite to be sensitized for a wide of applications.

### 3.3. Reactive Structures through Cast Curing

Cast curing is a well-known high volume production process in which the ingredients are incorporated into castable liquid polymer binder such as hydroxyl-terminated polybutadiene (HTPB), casted into specific size and shape in complex geometries, and cured to form macroscale solid composite structures [53]. Polyurethanes are the most widely used binder systems for cast curing of energetic materials because of the milder processing temperature of around 60 °C and better safety characteristics. In view of these advantages, cast curing based on polyurethane binder systems would be a suitable technique to safely process and mass produce macroscale reactive structures. However, it has not been widely exploited as a processing technique for fluoropolymer-based reactive structures due to the dearth of hydroxyl-terminated liquid prepolymers that are crucial for cast curing.

One known example of hydroxyl-terminated liquid fluorinated prepolymer is Fomblin^®^ ZDOL, a perfluoropolyalkylether (PFPAE) diol commercialized by Solvay [54] (Figure 14).

Fomblin^®^ ZDOL has been investigated as a fluorinated coating for nano-Al and its nanothermite compositions with the objective of inducing exothermic surface reaction between fluorine and the alumina shell, and thus promoting the overall reactivity [10,55] and dispersion in the polyurethane binder matrix [56].

Fluorinated polyurethanes derived from PFPAE diols have been reported to exhibit elastomeric behavior over a wide temperature range (between −75 °C and 100 °C), due to their multiphase morphology [57], which was ideal for binder application. However, they have not been used for the cast curing application of reactive materials. One plausible reason could be volatility issue with the PFPAE diols, that would lead to significant loss of the fluoropolymer before it can react with the metal fuel [58]. Therefore, it is necessary to develop cast curable fluoropolymers to adopt cast curing method for the processing of reactive structures.

At our institute, we developed a novel hydroxyl-terminated fluoropolymer (FP) with 58 wt% fluorine content for cast curing of nanothermites [59]. The liquid prepolymer was synthesized in single step process by the cationic ring-opening polymerization of commercially available 2,2,3,3,4,4,5,5,6,6,7,7,7-tridecafluoroheptyloxirane (TDFHO) fluoromonomer in the presence of 1,4-butanediol (BDO) initiator and boron trifluoride tetrahydrofuran complex catalyst (Figure 1).

The physicochemical properties of the synthesized FP were comparable to that of the prepolymers widely used for polyurethane binder applications for energetic materials [60]. The SNE composites respectively containing 50, 60 and 70 wt% Al/CuO nanothermite powders relative to FP were prepared by the cast curing process using dibutyltin dilaurate (DBTDL) catalyst, isophorone diisocyanate (IPDI) curing agent, and 2:1 mixture of BDO/trimethylolpropane (TMP) as chain extender (Figure 15).

Remarkably, the cast curing process involved only mild temperature conditions that never exceeded 60 °C at any point of time. As shown in the representative photographs (Figure 16), all the composite samples could be easily retrieved from the mold cured with minimal shrinkage and had smooth and shining surfaces.

Notably, the size of the casted composite specimen was in the centimeter scale, which was challenging to be achieved through the melt extrusion process. Furthermore, the viscosity of FP was remarkably low enough to accommodate the loading of a maximum of 70 wt% nanothermite with respect to the FP and successfully cast the curing mix into a macroscale geometry as shown in Figure 16.

The composites were further characterized for ignitability, combustion reactivity, and combustion temperature. All the cast cured nanothermite composites were readily ignitable by heated nichrome wire, even under an argon environment. A comparable formulation with inert HTPB binder did not ignite under similar conditions, and thus demonstrating the capability of the FP binder to induce pre-ignition reaction. The composite containing 60 wt% nanothermite and 40 wt% FP recorded a relatively higher heat of reaction of 880 ± 6 cal/g indicating that this formulation was closer to the stoichiometric condition. The 60 wt% nanothermite composites, due to the nearly stoichiometric but slightly fuel rich nature of the formulation, recorded the highest combustion reactivity. This was aided by an optimal contribution of fluorination and oxidation reactions as confirmed by the X-ray diffraction analysis of the combustion products.

The combustion temperature was determined with the help of emission spectroscopy technique after igniting the 60 wt% nanothermite composite sample using a 20W laser (Figure 17a). The Al/CuO nanothermite formulated at the equivalence ratio (ER = 2.0) was also tested as a control sample.

The acquired spectra of the samples are shown in Figure 17b. Emission spectra indicated that the nanothermite combustion occurs in the gas phase driven by the oxidation reaction of nano-Al by CuO, whereas that of the FP/nanothermite composite happens in the condensed phase through a combination of oxidation and fluorination reactions. As shown in Figure 17c, the 60 wt% nanothermite composite recorded an average combustion temperature of 2530 ± 134 K that was merely 14% lower than that recorded for the nanothermite alone (2950 ± 25 K). Even though further research needs to be conducted on the mechanical properties of the composites, this work represented a promising advance towards the preparation of larger scale reactive structures through cast curing.

In another strategy, fluorinated co-polyurethanes intended for cast curing applications of reactive materials were prepared by modifying existing prepolymers, such as GAP or HTPB with fluorinated segments. Zhang et al. [61] modified the poly(glycidyl azide-co-tetramethylene glycol) (PGT) prepolymer with a fluorinated segment (prop-2-yn-1-yl 4,4,5,5,6,6,6-heptafluorohexanoate) through a catalyst free azide-alkyne reaction, that was further simultaneously reacted with the IPDI and N-100 curing reagents and various amount of nano-Al to form nano-Al/PU reactive composites (Figure 2).

The 20 wt% nano-Al containing reactive composite recorded an exothermic heat of decomposition of 2.08 kJ/g, glass transition temperature of −28.1 °C, tensile strength of 2.1 ± 0.1 MPa, and elongation at break 477 ± 4%.

Tang et al. [62] reported a one pot method to prepare GAP-based fluorinated segmented polyurethane (FPU) binder by reacting a fluorinated diol (2,2,3,3-tetrafluoro-1,4-butanediol) chain extender with a pre-polymerized dicyclohexylmethylmethane-4,4’-diisocyanate (HMDI) terminated GAP (Figure 3).

As shown in Figure 3, the FPU consisted of the GAP soft segment and hard segment composed of the HMDI diisocyanate and the fluorinated diol chain extender. It is well established that the mechanical strength of the FPU is dependent on the relative changes in the hard and soft segment contents that would induce the desired micro phase separation and physical crosslinks. In this work, the hard segment content of the FPU was varied through changing the amount of the fluorinated diol used in the synthesis. The hard segment content was varied from 30 to 50 wt% with respect to the FPU that was formulated at an isocyanate to OH (*R*) ratio of 0.98. The number average molecular weight of the FPU ranged from 20,600 to 34,000 g/mol with PDI values from 2.34 to 2.48. With respect to increasing the hard segment content from 30 to 50 wt%, the tensile strength of the binder increased by 600% from ~1 MPa to ~7 MPa, and the strain at break decreased by 70% from ~2500% to 750%. The enhancement in the tensile strength of the FPU was attributed to the improved micro-phase separation, whereas the corresponding dilution of the soft segment content responsible for the elasticity resulted in the reduction in the elongation at break.

The FPUs also recorded a satisfactory thermal stability with onset temperature of the decomposition of ~210 °C and glass transition temperatures in the range of −32 to −35 °C. The increase in hard segment content also increased the fluorine element content of the FPUs from ~4.6 to 8.5 wt%. The heat of explosion of the binders measured by thermal ignition under an argon atmosphere recorded a decrease from 2139 ± 52 to 1643 ± 18 kJ/kg with respect to increasing hard segment content due to the inert nature of the hard segments.

The FPUs were further processed into reactive composite sheets containing 20 wt% of nano-Al using solvent casting followed by double-roll calendaring process and were characterized for the combustion efficiency and heat of explosion [63]. It was reported that the FPU/nano-Al composites exhibited a significantly higher combustion efficiency as compared to that without fluorine, resulting in the complete consumption of the nano-Al. Remarkably, despite the low fluorine content of ~3.7%, the FPU/nano-Al composite with 30 wt% hard segment recorded ~91% higher heat of explosion compared to the control composite without fluorine.

Xu et al. [64] reported an alternate type of GAP-based fluorinated polyurethane, synthesized by the copolymerization of GAP and 3,3-bis(2,2,2-trifluoroethoxymethyl) oxetane] glycol (BFMO) using toluene diisocyanate (TDI) as the crosslinker (Figure 4).

Three different compositions containing varying mole ratios of PBFMO and GAP (1:3, 1:9, and 1:19) with average molecular weights in the range of 33,000 g/mol were synthesized. The fluorine content of the polymers ranged 10.1% for 1:3 copolymer to 2% for the 1:19 copolymer. The tensile strength of the binder increased by ~50% from 2.9 ± 0.11 MPa to 5.75 ± 0.275 MPa with increasing fluorine content in the binder, whereas the elongation at break decreased by 24% from 2056 ± 47.3% to 1660 ± 42.3%. The improvement in the tensile strength was attributed to the improved phase separation of the hard and soft segments and enhancement in crosslinking density. Slow cook-off tests indicated that the Al composite of the 1:3 PBFMO-GAP binder (Al content not mentioned) was significantly more reactive than the control GAP/Al composite, which further demonstrated the higher energetic characteristic of Al-fluorine reaction.

In an earlier work, the same group had reported the synthesis and characterization of a cure castable fluorinated GAP [65], poly(TFEE-GAP) that was synthesized through an initial cationic copolymerization of epichlorohydrin (ECH) and 2,2,2-trifluoro-ethoxymethyl epoxy (TFEE), followed by azidation (Figure 5).

Poly(TFEE-GAP) had a glass transition temperature of −49.5 °C. The copolymer was converted into a copolyurethane network by using IPDI as the crosslinking agent and TMP as the chain extender and characterized for the mechanical properties. The fluorinated GAP copolyurethane network recorded a tensile strength of 5.52 MPa and elongation at break of 162.8% that was better than the corresponding control GAP polyurethane network.

## 4. Future Directions

As summarized in this review, fluoropolymer-based binder development for structural reactive material applications has become a niche area of research in the last decade. These research efforts are predominantly motivated by the necessity to obtain safely and economically processable alternatives to the difficult to process PTFE binder-based formulations. An alternative fluoropolymer binder candidate that enables the safe and high throughput processing of reactive material formulations as well as having matching oxidizing and mechanical properties of PTFE is yet to be developed. Research efforts in our institute and elsewhere have been focused on achieving this objective.

Although the adoption of additive manufacturing techniques is poised to make some inroads into the processing of non-PTFE fluoropolymer-based reactive structures into complex shapes, the key challenge to resolve is the limitation in the sizes of the reactive structures obtained. The solution to this challenge, as demonstrated in the examples in this review, will lie in the design of new low melting and cast curable fluoropolymers. This would render the corresponding reactive material formulations amenable to the high throughput melt processing/cast curing techniques prevalent in the energetic materials industry.

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
