# Peer review of "Nanoenergetic Composites with Fluoropolymers: Transition from Powders to Structures"

_molecules, 2022, doi:10.3390/molecules27196598_

Round 1

Reviewer 1 Report

This review paper summarized recent developments of nanoenergetic composites with fluoropolymers. The content is well organized. It provides a comprehensive and easy understanding description of the topic. There are only a few minor comments:

1) The first paragraph gives a bigger picture of this research field. It is better to cite some previous reviews and milestone articles here. 

2) Section 2 explains nicely why fluoropolymer would make a significant difference. However, the authors should also provide experimental results that directly support this idea. 

3) The production of fluoropolymers and decomposition cause fluorinated hazardous substances. Is this a concern? How to cope with this problem? 

Author Response

Thank you very much for the comments, which will certainly improve the quality of this review. Responses to the reviewer's comments.

1) The first paragraph gives a bigger picture of this research field. It is better to cite some previous reviews and milestone articles here. 

Ans: New references have been included in the introduction section. The newly added references are [6] Amreduri Macromol. Chem. Phys. 2020221, 1900573, [12] J.B.Delisio et.al; J. Phys. Chem. B , 120, 24, 5534–5542 (2016)[13] J.Chen et.al;  Mater. Res. Express 7 115009 (2020) [14] F.D.Ruz-Nuglo et.al; Adv. Eng. Mater., 20(2), 1700390 (2018)

2. Section 2 explains nicely why fluoropolymer would make a significant difference. However, the authors should also provide experimental results that directly support this idea. 

 Ans: The activation effect of the fluoropolymers on  the combustion of Al and Al-Mg alloy powders was directly confirmed in the research work at our institute.  The results were published in H.Nie et.al; Combustion and Flame 220 (2020) 394–406 and has been  added to the section 2 [ln 133 to 138; Page 2] of the review.

3) The production of fluoropolymers and decomposition cause fluorinated hazardous substances. Is this a concern? How to cope with this problem? 

Yes, we fully agree with the reviewer on the hazard associated with the production of fluroopolymer. It is certainly of high concern for  human and environmental health. In academic settings, since the quantity of the fluoropolymer being used is comparatively less, the problem could be coped up with effective implementation of engineering controls as well as through the use of PPEs while working with the fluoropolymer based compositions. 

Reviewer 2 Report

Nano-energetic materials have been widely use in the defence and aerospace field. The ignition and combustion features of nano-energetic materials is becoming a hotspot issue in the field of energetic or reactive materials. Thus, it’s very good to find such a review focuses on the major research developments of polymer bonded nanoenergetic materials. The authors systematically reviewed the research on ignitability and reactive structures of fluoropolymer/nano aluminum oxidation composites.

I have the following comments which need the authors consider to supplement:

(1)    In line 152, the authors indicated that impact initiation potential of the composite is closely related to the dynamic deformation behavior of the composites. How do the authors think the ignition or reactive mechanisms of the composite under impact loading? As we know, the input energy resulting from impact loading is hard to produce a sufficient bulk temperature rise, thus prohibiting ignition and reaction. Are there some hotspot mechanisms like polymer bonded explosives?

(2)    In line 171, please add more statements on the mechanisms that how do thermal softening of the PTFE matrix influence critical ignition hotspots. As the decrease of stress level of matrix, the mechanical work rate is generally decreasing so that the localized hotspot maybe not easy to reach a high temperature.

(3)    How do the authors think the effects of the interface between Al and PTFE on its ignition and combustion properties? If there is a debonding of interface, the surface area will increase and might remarkably influence the combustion performance.

(4)    What the difference of melt extrusion, solvent casting, and cast curing in the manufacturing? The authors should provide a brief introduction to help readers in the related field to understand.

(5)    In line 586, the authors said that the AlFA composite could be ignited at a very low impact velocity of 35 m/s by the addition of 15% glass micro-balloons. What is the reason for that behavior?

Author Response

Thank you for providing comments for the manuscript which will help to improve the quality. Our response to the comments are given below.

1. In line 152, the authors indicated that impact initiation potential of the composite is closely related to the dynamic deformation behavior of the composites. How do the authors think the ignition or reactive mechanisms of the composite under impact loading? As we know, the input energy resulting from impact loading is hard to produce a sufficient bulk temperature rise, thus prohibiting ignition and reaction. Are there some hotspot mechanisms like polymer bonded explosives?

Ans: Impact induced ignition in PTFE/Al composites is plausibly driven by shear induced mechanism as proposed by  Ames [R.G.Ames, Energy Release Characteristics of Impact-Initiated Energetic Materials;Mater. Res. Soc. Symp. Proc. Vol. 896 (2006);]. As a result of the thermal softening of the bulk material due to the influence of the high shear stress during impact, adiabating shear banding will occur. Adiabatic shear banding, a well known mechanism for deposition of energy, creates localized hot spots comparable in principle to that of thermally induced hotspots resulting in ignition and reaction.

2. In line 171, please add more statements on the mechanisms that how do thermal softening of the PTFE matrix influence critical ignition hotspots. As the decrease of stress level of matrix, the mechanical work rate is generally decreasing so that the localized hotspot maybe not easy to reach a high temperature.

Ans:Mechanistically, the thermal softening of PTFE leads to the  formation of the adiabatic shear bands that generates the  critical  hotspots responsible for the ignition of the composite  . A statement and  a relevance reference to this effect has been added to the manuscript.

3. How do the authors think the effects of the interface between Al and PTFE on its ignition and combustion properties? If there is a debonding of interface, the surface area will increase and might remarkably influence the combustion performance.

Ans: We think that the debonding of the interface is a critical phenomenon that would remarkably influence the combustion performance. It is well known that, shear failure is one of the several mechanisms that are capable of producing hot spots for ignition. The shear failure is primarily driven by the interfacial debonding between the PTFE binder and Al particles. Thus interfacial debonding is a key mechanism promoting the ignition. Once the failure occurs, the exposed surface area would promote the diffusion controlled reaction [E.M.Hunt et.al; Impact ignition of nano and micron composite energetic materials. Int. J. Impact Eng, 36, 842 (2009).] between the Al and the oxidizer thus resulting in the propagation of the combustion.

4. What the difference of melt extrusion, solvent casting, and cast curing in the manufacturing? The authors should provide a brief introduction to help readers in the related field to understand.

Ans:The differences among the manufacturing operations have been briefly elaborated as follows and it has been included in the manuscript [Page 7; lines 251 to 270].

"In melt extrusion, the molten thermoplastic binder and the reactive ingredients are extruded with a rotating screw under elevated temperature through a die into a structure of uniform shape. The intense mixing of the screw assists in the uniform dispersion of the ingredients in the polymer matrix. Melt extrusion operates continuously thereby reducing the number of involved unit operations and improves the economic viability of the reactive structure manufacturing. Nevertheless, the quality and performance of reactive structures processed through melt extrusion is limited by the friction and thermal sensitivities of the reactive ingredients in the formulation. The solvent casting process as the name implies uses a binder soluble in a volatile solvent to disperse the reactive ingredients at a reasonable solid concentration. The dispersion is casted into a mold and the solvent evaporated to obtain the requisite shape. Viscosity of the dispersion is a critical parameter determining the quality, size, and performance of the reactive structure. A key challenge is the residual porosity in the structures caused by evaporation of the solvent that requires to be tackled by the application of vacuum. In cast curing, the reactive ingredients are mixed with a liquid prepolymer, cast into a mold and the mixture converted into a permanent polymeric binder bound composite by reacting with a suitable curing reagent. The most traditional cure chemistry employed is based upon polyurethane linkage formed by the reaction of the hydroxyl terminated liquid prepolymer and multifunctional isocyanate curing reagent. The key process parameters to be considered are a) ability of the binder to wet solids b) final mix viscosity, and c) flowability of the mix into the mold".

5. In line 586, the authors said that the AlFA composite could be ignited at a very low impact velocity of 35 m/s by the addition of 15% glass micro-balloons. What is the reason for that behavior?

The reason for this behaviour is the sensitization of the composite by adding the glass micro-balloons. The addition of the 15% glass micro-balloons  creates effective "pinch points" that would result in the viscous heating and consequent ignition of the composite. This statement has been added to the revised manuscript Page 18; line 636

Reviewer 3 Report

This is a well-written review paper reporting on nanoenergetic composites with fluoropolymers.  The review does not only cover fluoropolymers but also includes manufacturing / (3D) printing with a special focus on energetic materials.  The different reaction mechanisms between the oxidation of Al in the absence and presence of fluoropolymers are well discussed.

The review is ready for acceptance as it is.

Author Response

Thank you very much for reviewing this manuscript.